# PRIVATE GANS, REVISITED

## ABSTRACT

We show that with improved training, the standard approach for differentially private GANs – updating the discriminator with noisy gradients – achieves or competes with state-of-the-art results for private image synthesis. Existing instantiations of this approach neglect to consider how adding noise *only to discriminator updates* disrupts the careful balance between generator and discriminator necessary for successful GAN training. We show that a simple fix – taking more discriminator steps between generator steps – restores parity and improves training. Furthermore, with the goal of restoring parity between the generator and discriminator, we experiment with further modifications to improve discriminator training and see further improvements in generation quality. For MNIST at $\varepsilon = 10$, our private GANs improve the record FID from $48.4$ to $13.0$, and record downstream classifier accuracy from $83.2\%$ to $95.0\%$.

## 1 INTRODUCTION

Differential privacy (DP) (Dwork et al., 2006b) has emerged as a compelling approach for training machine learning models on sensitive data. However, incorporating DP requires significant changes to the training process. Notably, it prevents the modeller from working directly with private data, complicating debugging and exploration. Furthermore, the modeller can no longer interact with a private dataset after exhausting their allocated privacy budget. One approach to alleviate these issues is by producing *differentially private synthetic data*, which can be plugged directly into existing machine learning pipelines, without further concern for privacy.

A recent line of work studies leveraging deep generative models to produce DP synthetic data. Early efforts focused on privatizing generative adversarial networks (GANs) (Goodfellow et al., 2014) by using differentially private stochastic gradient descent (DPSGD) (Abadi et al., 2016) to update the GAN discriminator – an approach referred to as *DPGAN* (Xie et al., 2018; Beaulieu-Jones et al., 2019; Torkzadehmahani et al., 2019).

However, follow-up work has significantly departed from this baseline DPGAN approach, either in terms of: (a) *the privatization scheme*, in favor of approaches based on subsample-and-aggregate which divide the data into $\geq 1000$ disjoint partitions and train teacher discriminators separately on each one (Jordon et al., 2019; Long et al., 2021; Chen et al., 2020; Wang et al., 2021); or (b) *the generative modelling framework altogether*, opting instead to minimize notions of statistical distance between real and generated data, such as maximum mean discrepancy (Harder et al., 2021; Vinaroz et al., 2022), or Sinkhorn divergences (Cao et al., 2021).

For labelled image synthesis, these custom generative models designed specifically for privacy fall short of GANs when evaluated at their non-private limits ($\varepsilon \to \infty$), suggesting limited scalability to larger, higher-resolution datasets.[1] On the other hand, the literature corroborates that under modest privacy budgets, these departures from the baseline DPGAN lead to significant improvements in generation quality. Proposed explanations attribute these results to inherent limitations of the DPGAN framework, suggesting that either: (a) privatizing discriminator training is sufficient for privacy, but may be overkill when only the generator needs to be released (Long et al., 2021); or (b) adversarial objectives may be unsuited for training under privacy (Cao et al., 2021).

---

[1]For example, the record FID for MNIST at $\varepsilon = 10$ is $48.4$ (Cao et al., 2021). When evaluated at $\varepsilon = \infty$, their method achieves an FID of $43.4$. Our non-private GANs obtain an FID of $3.2$.

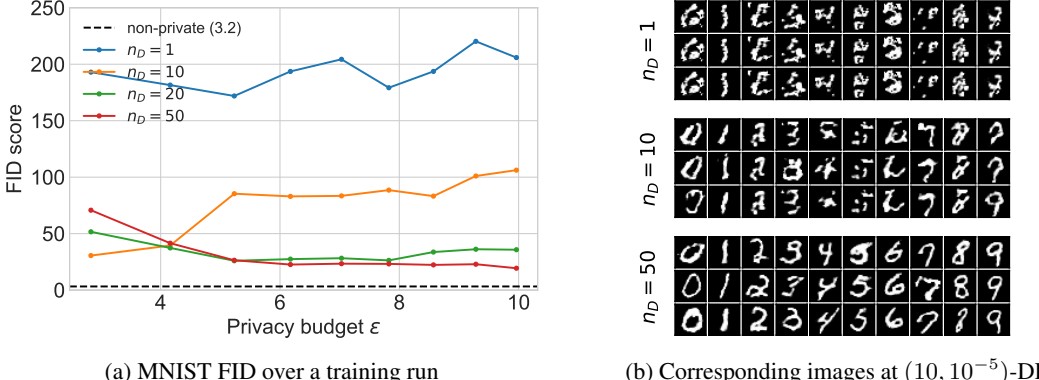

(a) MNIST FID over a training run      (b) Corresponding images at $(10, 10^{-5})$-DP

Figure 1: DPGAN results on MNIST synthesis at $(10, 10^{-5})$-DP. **(a)** We find that increasing $n_{\mathcal{D}}$, the number of discriminator steps taken between generator steps, significantly improves image synthesis results. Using $n_{\mathcal{D}} = 50$ instead of $n_{\mathcal{D}} = 1$ improves FID from $205.9 \rightarrow 19.4$, which improves over the record FID of $48.4$ from Cao et al. (2021). $n_{\mathcal{D}} = 50$ also improves the record downstream classification accuracy to $92.9\%$ (see Figure 2a), an improvement over the record accuracy of $83.2\%$ from Cao et al. (2021). **(b)** Corresponding synthesized images. We observe that large $n_{\mathcal{D}}$ improves visual quality, and low $n_{\mathcal{D}}$ leads to mode collapse.

**Our contributions.** We demonstrate that the reported poor results of DPGANs should not be attributed to inherent limitations of the framework, but rather, training issues. Specifically, we propose that the *asymmetric noise addition* in DPGANs (adding noise to discriminator updates only) weakens the discriminator relative to the generator, disrupting the careful balance necessary for successful GAN training. We propose that taking more discriminator steps between generator updates addresses the imbalance introduced by noise. With this change, DPGANs improve significantly (see Figure 1), going from non-competitive to achieving or competing with state-of-the-art results in private image synthesis.

Furthermore, we show this perspective on private GAN training ("restoring parity to a discriminator weakened by DP noise") can be applied to improve training. We make other modifications to discriminator training – large batch sizes and adaptive discriminator step frequency – to further improve upon the aforementioned results.

In summary, we make the following contributions:

1. We find that taking more discriminator steps between generator steps significantly improves DPGANs. Contrary to the previous results in the literature, DPGANs *do* compete with state-of-the-art generative modelling approaches designed with privacy in mind.

2. We present empirical findings towards understanding why more frequent discriminator steps help. We propose an explanation based on *asymmetric noise addition* for why vanilla DPGANs do not perform well, and why taking more steps helps.

3. We put our explanation to the test. We employ it as a principle for designing better private GAN training recipes, and indeed are able to improve over the aforementioned results.

## 2   Preliminaries

Our goal is to train a generative model on sensitive data that is safe to release, i.e., it does not leak the secrets of individuals in the training dataset. We do this by ensuring the training algorithm $\mathcal{A}$ – which takes as input the sensitive dataset $D \in \mathcal{U}$ and returns the parameters of a trained (generative) model $\theta \in \Theta$ – satisfies differential privacy.

**Definition 1** (Differential Privacy (Dwork et al., 2006b)). A randomized algorithm $\mathcal{A} : \mathcal{U} \rightarrow \Theta$ is $(\varepsilon, \delta)$-*differentially private* if for every pair of neighbouring datasets $D, D' \in \mathcal{U}$, we have

$$\mathbb{P}\{\mathcal{A}(D) \in S\} \leq \exp(\varepsilon) \cdot \mathbb{P}\{\mathcal{A}(D') \in S\} + \delta \qquad \text{for all } S \subseteq \Theta.$$

In this work, we adopt the add/remove definition of DP, and say two datasets $D$ and $D'$ are neighbouring if they differ in at most one entry, that is, $D = D' \cup \{x\}$ or $D' = D \cup \{x\}$.

---

**Algorithm 1** TrainDPGAN($D; \cdot$)

---

1: **Input:** Labelled dataset $D = \{(x_j, y_j)\}_{j=1}^n$. Discriminator $\mathcal{D}$ and generator $\mathcal{G}$ initializations $\phi_0$ and $\theta_0$. Optimizers OptD, OptG. Privacy parameter $\delta$. Hyperparameters: $n_\mathcal{D}$ ($\mathcal{D}$ steps per $\mathcal{G}$ step), $T$ (total number of $\mathcal{D}$ steps), $B$ (expected batch size), $C$ (clipping norm), $\sigma$ (noise multiplier).
2: $q \leftarrow B/|D|$ and $t, k \leftarrow 0$ ▷ Calculate sampling rate $q$, initialize counters.
3: **while** $t < T$ **do** ▷ Update $\mathcal{D}$ with DPSGD.
4: $\quad S_t \sim \text{PoissonSample}(D, q)$ ▷ Sample a real batch $S_t$ by including each $(x, y) \in D$ w.p. $q$.
5: $\quad \widetilde{S}_t \sim \mathcal{G}(\cdot; \theta_k)^B$ ▷ Sample fake batch $\widetilde{S}_t$.
6: $\quad g_{\phi_t} \leftarrow \sum_{(x,y) \in S_t} \text{clip}\left(\nabla_{\phi_t}(-\log(\mathcal{D}(x, y; \phi_t))); C\right)$
$\qquad + \sum_{(\widetilde{x},\widetilde{y}) \in \widetilde{S}_t} \text{clip}\left(\nabla_{\phi_t}(-\log(1 - \mathcal{D}(\widetilde{x}, \widetilde{y}; \phi_t))); C\right)$ ▷ Clip per-example gradients.
7: $\quad \widehat{g}_{\phi_t} \leftarrow \frac{1}{2B}(g_{\phi_t} + z_t)$, where $z_t \sim \mathcal{N}(0, C^2\sigma^2 I))$ ▷ Add Gaussian noise.
8: $\quad \phi_{t+1} \leftarrow \text{OptD}(\phi_t, \widehat{g}_{\theta_t})$ and $t \leftarrow t + 1$
9: $\quad$ **if** $n_\mathcal{D}$ divides $t$ **then** ▷ Perform $\mathcal{G}$ update every $n_\mathcal{D}$ steps.
10: $\qquad \widetilde{S}'_t \sim \mathcal{G}(\cdot; \theta_k)^B$
11: $\qquad g_{\theta_k} \leftarrow \frac{1}{B} \sum_{(\widetilde{x},\widetilde{y}) \in \widetilde{S}'_t} \nabla_{\theta_k}(-\log(\mathcal{D}(\widetilde{x}, \widetilde{y}; \phi_t)))$
12: $\qquad \theta_{k+1} \leftarrow \text{OptG}(\theta_k, g_{\theta_k})$ and $k \leftarrow k + 1$
13: $\quad$ **end if**
14: **end while**
15: $\varepsilon \leftarrow \text{PrivacyAccountant}(T, \sigma, q, \delta)$ ▷ Compute privacy budget spent.
16: **Output:** Final $\mathcal{G}$ parameters $\theta_k$. $(\varepsilon, \delta)$-DP guarantee.

---

We highlight one convenient property of DP, known as *closure under post-processing*. This says that interacting with a privatized model (e.g., using it to compute gradients on non-sensitive data, generate samples) does not lead to any further privacy violation.

**Proposition 2** (Post-processing). Let $\mathcal{A} : \mathcal{U} \to \Theta$ be a randomized algorithm that is $(\varepsilon, \delta)$-DP, and $f : \Theta \to \mathcal{Y}$ be an arbitrarily randomized mapping. Then $f \circ \mathcal{A} : \mathcal{U} \to \mathcal{Y}$ is $(\varepsilon, \delta)$-DP.

**DPSGD.** A gradient-based training algorithm can be privatized by employing *differentially private stochastic gradient descent* (DPSGD) (Song et al., 2013; Bassily et al., 2014; Abadi et al., 2016) as a drop-in replacement for SGD. DPSGD involves clipping per-example gradients and adding Gaussian noise to their sum, which effectively bounds and masks the contribution of any individual point to the final model parameters. Privacy analysis of DPSGD follows from several classic tools in the DP toolbox: Gaussian mechanism, privacy amplification by subsampling, and composition (Dwork et al., 2006a; Dwork & Roth, 2014; Abadi et al., 2016; Wang et al., 2019). Our work employs the DPSGD analysis of Mironov et al. (2019) implemented in Opacus (Yousefpour et al., 2021).

**DPGANs.** Algorithm 1 details the training algorithm for DPGANs, which is effectively an instantiation of DPSGD. Note that only gradients for the discriminator $\mathcal{D}$ must be privatized (via clipping and noise), and not those for the generator $\mathcal{G}$. This is a consequence of post-processing (Proposition 2) – the generator only interacts with the sensitive dataset indirectly via discriminator parameters, and therefore does not need further privatization.

## 3 FREQUENT DISCRIMINATOR STEPS IMPROVES PRIVATE GANS

In this section, we discuss our main finding: the number of discriminator steps taken between each generator step ($n_\mathcal{D}$ from Algorithm 1) plays a significant role in the success of private GAN training. For a fixed setting of DPSGD hyperparameters, there is an optimal range of values for $n_\mathcal{D}$ that maximizes generation quality, in terms of both visual quality and utility for downstream classifier training. This value is often quite large ($n_\mathcal{D} \approx 100$ in some cases).

### 3.1 EXPERIMENTAL DETAILS

**Setup.** We focus on labelled generation of MNIST (LeCun et al., 1998) and FashionMNIST (Xiao et al., 2017), both of which are comprised of $60000$ $28 \times 28$ grayscale images divided into 10 classes. To build a strong baseline, we begin from an open source PyTorch (Paszke et al., 2019) implemen-

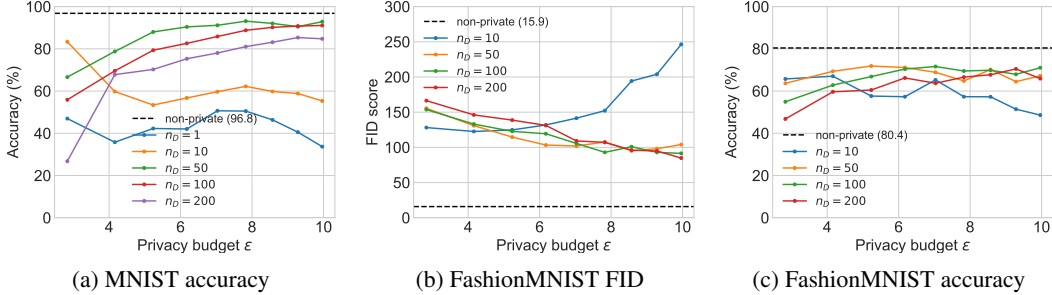

Figure 2: DPGAN results over training runs using different discriminator update frequencies $n_{\mathcal{D}}$, targeting $(10, 10^{-5})$-DP. **(a)** As a measure of synthetic data utility, we plot the test set accuracy of a CNN trained on generated data only. Accuracy mirrors FID scores (see Figure 1a). Going from $n_{\mathcal{D}} = 1$ to $n_{\mathcal{D}} = 50$ improves accuracy from $33.7\% \rightarrow 92.9\%$. Further increases hurts accuracy. **(b) & (c)** We obtain similar results for FashionMNIST. Note that the optimal $n_{\mathcal{D}}$ is higher (around $n_{\mathcal{D}} \approx 100$). At $n_{\mathcal{D}} = 100$, we obtain an FID of 91.5 and accuracy of 71.1%, which compares favourably to the record FID of 128.3 and record accuracy of 75.5% reported in Cao et al. (2021) for $(10, 10^{-5})$-DP generation of FashionMNIST.

tation[2] of DCGAN (Radford et al., 2016) that performs well non-privately, and copy their training recipe. We then adapt their architecture to our purposes: removing BatchNorm layers (which are not compatible with DPSGD) and adding label embedding layers to enable labelled generation. Training this configuration non-privately yields labelled generation that achieves FID scores of 3.2 on MNIST and 15.9 on FashionMNIST. Finally, we note that these models are not small: $\mathcal{D}$ and $\mathcal{G}$ have 1.72M and 2.27M trainable parameters respectively. Please see Appendix B.1 for more details.

**Privacy implementation.** To privatize training, we use Opacus (Yousefpour et al., 2021) which implements per-example gradient computation and the RDP accounting of Mironov et al. (2019). For our baseline setting, we use the following DPSGD hyperparameters: we keep the non-private (expected) batch size $B = 128$, and use a noise scale $\sigma = 1$ and clipping norm $C = 1$. Under these settings, we have the budget for $T = 450000$ discriminator steps when targeting $(10, 10^{-5})$-DP.

**Evaluation.** We evaluate our generative models by examining the *visual quality* and *utility for downstream tasks* of generated images. Following prior work, we measure visual quality by computing the Fréchet Inception Distance (FID) (Heusel et al., 2017) between 60000 generated images and entire test set.[3] To measure downstream task utility, we again follow prior work, and train a CNN classifier on 60000 generated image-label pairs and report its accuracy on the real test set.

## 3.2 RESULTS

**More frequent discriminator steps improves generation.** We plot in Figures 1a and 2 the evolution of FID and downstream accuracy during DPGAN training for both MNIST and FashionMNIST, under varying discriminator update frequencies $n_{\mathcal{D}}$. The effect of this parameter has outsized impact on the final results. For MNIST, $n_{\mathcal{D}} = 50$ yields the best results; on FashionMNIST, the best FID is obtained at $n_{\mathcal{D}} = 200$ and the best accuracy at $n_{\mathcal{D}} = 100$.

**Private GANs are on a path to mode collapse.** For the MNIST results in Figures 1a and 2a, we observe that at low discriminator update frequencies ($n_{\mathcal{D}} = 10$), the best FID and accuracy scores occur early in training, *well before the privacy budget we are targeting is exhausted*.[4] In fact, at 50000 discriminator steps ($\varepsilon \approx 2.85$), $n_{\mathcal{D}} = 10$ has better FID (30.6) and accuracy (83.3%) than other settings of $n_{\mathcal{D}}$. However, these results deteriorate with continued training. In Figure 3, we

---

[2]Courtesy of Hyeonwoo Kang (https://github.com/znxlwm). Code available at this link.

[3]We use an open source PyTorch implementation to compute FID: https://github.com/mseitzer/pytorch-fid.

[4]This observation has been reported in (Neunhoeffer et al., 2021), serving as motivation for their remedy of taking a mixture of intermediate models encountered in training. We are not aware of any mentions of this aspect of DPGAN training in papers reporting DPGAN baselines for labelled image synthesis.

$$t = 50\text{K} \qquad t = 100\text{K} \qquad t = 150\text{K} \qquad t = 200\text{K}$$

Figure 3: Evolution of samples drawn during training with $n_{\mathcal{D}} = 10$, when targeting $(10, 10^{-5})$-DP. This setting reports its best FID and downstream accuracy at $t = 50\text{K}$ iterations ($\varepsilon \approx 2.85$). As training progresses, we observe mode collapse for several classes alongside the deterioration in evaluation metrics.

plot the evolution of generated images for this $n_{\mathcal{D}} = 10$ run over the course of training, and observe qualitative evidence of mode collapse, *co-occurring* with the deterioration in FID and accuracy.

**An optimal discriminator update frequency.** These results suggest that *fixing other DPSGD hyperparameters, there is an optimal setting for the discriminator step frequency* $n_{\mathcal{D}}$ that strikes a balance between: (1) being too low, causing the generation quality to peak early in training and then undergo mode collapse; resulting in all subsequent training to consume additional privacy budget *without improving the model*; and (2) being too high, preventing the generator from taking enough steps to converge before the privacy budget is exhausted (an example of this is the $n_{\mathcal{D}} = 200$ run in Figure 2a). Striking this balance results in the most effective utilization of privacy budget towards improving the generator.

## 4 WHY DOES TAKING MORE STEPS HELP?

In this section, we present empirical findings towards understanding why more frequent discriminator steps improves DPGAN training. We propose an explanation that is conistent with our findings.

**How does DP affect GAN training?** Figure 4 compares the accuracy of the GAN discriminator (on held-out real and fake examples) immediately before each generator step between non-private training and private training with different settings of $n_{\mathcal{D}}$. We observe that non-privately, discriminator accuracy stays around 60% throughout training. Naively introducing DP ($n_{\mathcal{D}} = 1$) leads to a qualitative difference: DP causes discriminator accuracy to drop to 50% immediately at the start of training, and never recovers.[5]

For other settings of $n_{\mathcal{D}}$, we make three observations: (1) larger $n_{\mathcal{D}}$ corresponds to higher accuracy; (2) the generator improves during the periods in which the discriminator stays above 50% accuracy; and (3) accuracy decreases throughout training as the generator improves, and degradation/stagnation of the generator (as observed in Figure 3) co-occurs with discriminator accuracy dropping to 50%.

Based on these observations, we propose the following explanation for why more steps help:

- Generator improvement occurs when the discriminator is capable of distinguishing between real and fake data.
- The *asymmetric noise addition* introduced by DP to the discriminator makes such a task difficult, resulting in limited generator improvement.
- Allowing the discriminator to train longer on a fixed generator improves its accuracy, recovering the non-private case where the generator and discriminator are balanced.

**Does reducing noise accomplish the same thing?** In light of this explanation, we ask if reducing the noise level $\sigma$ can offer the same improvement as taking more steps, as reducing $\sigma$ should also improve discriminator accuracy before a generator step. To test this: starting from our setting in Section 3, fixing $n_{\mathcal{D}} = 1$, and targeting MNIST at $\varepsilon = 10$, we search over a grid of noise levels

---

[5]Our plot only shows the first 20000 generator steps, but we remark that this persists until the end of training (450000 steps).

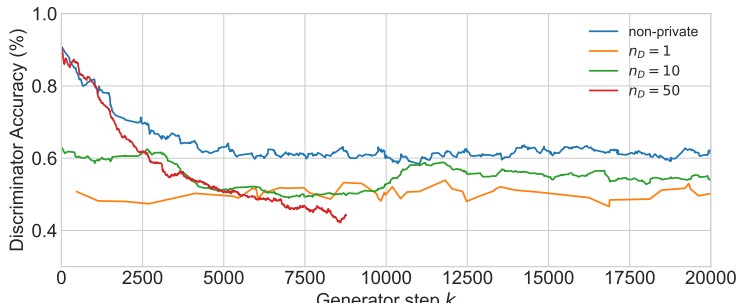

Figure 4: Discriminator accuracy immediately before each generator step. While non-privately the discriminator maintains a 60% accuracy, the private discriminator with $n_\mathcal{D} = 1$ is effectively a random guess. Increasing the number of discriminator steps recovers the discriminator's advantage early on, leading to generator improvement. An improved generator makes the discriminator's task more difficult, driving down accuracy.

$\sigma = \{0.4, 0.43, 0.45, 0.5, 0.55, 0.6, 0.7, 0.8\}$; the lowest of which, $\sigma = 0.4$, admits a budget of only $T = 360$ discriminator steps. We obtain a best FID of $127.1$ and best accuracy of $57.5\%$ at noise level $\sigma = 0.45$. Hence we can conclude that in this setting, incorporating discriminator update frequency in our design space allows for more effective use of privacy budget for improving the discriminator, and in turn, generation quality.

**Does taking more discriminator steps always help?** As we discuss in more detail in Section 5.1, when we are able to find other means to improve the discriminator beyond taking more steps, tuning discriminator update frequency may not yield improvements. To illustrate with an extreme case, consider eliminating the privacy constraint. In non-private GAN training, taking more steps is known to be unnecessary. We corroborate this result: we run our non-private baseline from Section 3 with the same number of generator steps, but opt to take 10 discriminator steps between each generator step instead of 1. FID worsens from $3.2 \rightarrow 8.3$, and accuracy worsens from $96.8\% \rightarrow 91.3\%$.

## 5 BETTER GENERATORS VIA BETTER DISCRIMINATORS

Our proposed explanation in Section 4 provides a concrete suggestion for improving GAN training: effectively use our privacy budget to maximize the number of generator steps taken when the discriminator has sufficiently high accuracy. We experiment with modifications to the private GAN training recipe towards these ends, which translate to improved generation.

### 5.1 LARGER BATCH SIZES

Several recent works have demonstrated that for classification tasks, DPSGD achieves higher accuracy with larger batch sizes, after tuning the noise scale $\sigma$ accordingly (Tramèr & Boneh, 2021; Anil et al., 2021; De et al., 2022). GAN training is typically conducted with small batch sizes (for example, DCGAN uses $B = 128$, which we adopt; StyleGAN uses $B = 32$). Therefore it is interesting to see if large batch sizes indeed improve private GAN training. We corroborate that larger batch sizes do not significantly improve our non-private MNIST baseline from Section 3: when we go up to $B = 2048$ from $B = 128$, FID stays at $3.2$ and accuracy improves from $96.8\% \rightarrow 97.5\%$.

**Results.** We scale up batch sizes, considering $B \in \{64, 128, 512, 2048\}$, and search for the optimal noise scale $\sigma$ and $n_\mathcal{D}$ (details in Appendix B.2). We target both $\varepsilon = 1$ and $\varepsilon = 10$. We report the best results from our hyperparameter search in in Table 1. We find that larger batch sizes leads to improvements: for $\varepsilon = 10$, the best MNIST and FashionMNIST results are achieved at $B = 2048$. For $\varepsilon = 1$, the best results are achieved at $B = 512$. We also note that for large batch sizes, the optimal number of generator steps can be quite small. For $B = 2048$, $\sigma = 4.0$, targeting MNIST at $\varepsilon = 10$, $n_\mathcal{D} = 5$ is the optimal discriminator update frequency, and improves over our best $B = 128$ setting employing $n_\mathcal{D} = 50$.

## 5.2 Adaptive discriminator step frequency

Our observations from Section 3 and 4 motivate us to consider *adaptive* discriminator step frequencies. As pictured in Figure 4, discriminator accuracy drops during training as the generator improves. In this scenario, we want to take more steps to improve the discriminator, in order to further improve the generator. However, using a large discriminator update frequency right from the beginning of training is wasteful – as evidenced by the fact that low $n_\mathcal{D}$ achieves the best FID and accuracy early in training. Hence we propose to start at a low discriminator update frequency ($n_\mathcal{D} = 1$), and ramp up when our discriminator is performing poorly.

Accuracy on real data must be released with DP. While this is feasible, it introduces the additional problem of having to find the right split of privacy budget for the best performance. We observe that discriminator accuracy is related to discriminator accuracy on fake samples only (which are free to evaluate on, by post-processing). Hence we use it as a proxy to assess discriminator performance.

The adaptive step frequency is parameterized by two terms, $\beta$ and $d$. $\beta$ is the decay parameter used to compute the exponential moving average (EMA) of discriminator accuracy on fake batches before each generator update. We use $\beta = 0.99$ in all settings. $d$ is the accuracy floor that upon reaching, we move to the next update frequency $n_\mathcal{D} \in \{1, 2, 5, 10, 20, 50, 100, 200, 500\}$. We try $d = 0.6$ and $d = 0.7$, finding that $0.7$ works better for large batches. Additionally, we promise a grace period of $2/(1 - \beta) = 200$ generator steps before moving on to the next update frequency. This formula is motivated by the fact that $\beta$-EMA's value is primarily determined by its last $2/(1 - \beta)$ observations.

The additional benefit of the adaptive step frequency is that it means we do not have to search for the optimal update frequency. Although the adaptive step frequency introduces the extra hyperparameter of the threshold $d$, we found that these two settings ($d = 0.6$ and $d = 0.7$) were sufficient to improve over results of a much more extensive hyperparameter search.

## 5.3 Comparison with previous results in the literature

### 5.3.1 MNIST and FashionMNIST

Table 1 summarizes our best experimental settings for MNIST and FashionMNIST, and situates them in the context of previously reported results for the task. We provide some example generated images in Figures 7 and 8 for $\varepsilon = 10$, and Figures 9 and 10 for $\varepsilon = 1$.

**Simple DPSGD beats all alternative GAN privatization schemes.** Our baseline DPGAN from Section 3, with the appropriate choice of $n_\mathcal{D}$ (and without the modifications described in this section yet), outperforms all other GAN-based approaches proposed in the literature (GS-WGAN, PATE-GAN, G-PATE, and DataLens) *uniformly* across both metrics, both datasets, and both privacy levels.

**Large batch sizes and adaptive step schedules improve GAN training.** Broadly speaking, across both privacy levels and both datasets, we see an improvement from taking larger batch sizes, and then another with an adaptive step schedule. The magnitude of improvement varies.

**Comparison with state-of-the-art.** In the low privacy/high $\varepsilon$ regime, most of our results are dramatically better than prior work[6] – for example, decreasing FID from $48.4$ to $13.0$ and increasing accuracy from $83.2\%$ to $95.0\%$ on MNIST. In the high privacy/low $\varepsilon$ regime, improvements are not quite as extreme, but can still be significant (FID for MNIST and FashionMNIST), and only compare negatively to state-of-the-art for accuracy on FashionMNIST. Visual comparison for $\varepsilon = 10$ results in 5

### 5.3.2 CelebA-Gender

We also report results on generating $32 \times 32$ CelebA, conditioned on gender at $(10, 10^{-6})$-DP. For these experiments, we used slightly larger models (2.64M and 3.16M parameters for $\mathcal{D}$ and $\mathcal{G}$

---

[6]We do not compare with two recent works on private generative models (Chen et al., 2022; Jiang et al., 2022), as we believe there are gaps in their privacy analyses. This has been confirmed by the authors of Jiang et al. (2022), and the sketch of an argument regarding non-privacy of Chen et al. (2022) has been shared with us by others (Anonymous, 2022).

| Privacy Level | Method | Reported In | MNIST | | FashionMNIST | |
|---|---|---|---|---|---|---|
| | | | FID | Acc.(%) | FID | Acc.(%) |
| $\varepsilon = \infty$ | Real data | (This work) | 1.0 | 99.2 | 1.5 | 92.5 |
| | GAN | | 3.2 | 96.8 | 15.9 | 80.4 |
| | DPGAN[7] | Chen et al. (2020) | 179.16 | 63 | 243.80 | 50 |
| | | Long et al. (2021) | 304.86 | 80.11 | 433.38 | 60.98 |
| | GS-WGAN | Chen et al. (2020) | 61.34 | 80 | 131.34 | 65 |
| | PATE-GAN | Long et al. (2021) | 253.55 | 66.67 | 229.25 | 62.18 |
| $\varepsilon = 10$ | G-PATE | Long et al. (2021) | 150.62 | 80.92 | 171.90 | 69.34 |
| | DataLens | Wang et al. (2021) | 173.50 | 80.66 | 167.68 | 70.61 |
| | DP-MERF | Cao et al. (2021) | 116.3 | 82.1 | 132.6 | **75.5** |
| | DP-Sinkhorn | Cao et al. (2021) | 48.4 | 83.2 | 128.3 | 75.1 |
| | DPGAN | | 19.4 | 92.9 | 91.5 | 71.1 |
| | + large batches | (This work) | 13.2 | 94.3 | 66.7 | 72.1 |
| | + step schedule | | **13.0** | **95.0** | **56.8** | 74.8 |
| | DPGAN | Long et al. (2021) | 470.20 | 40.36 | 472.03 | 10.53 |
| | GS-WGAN | Long et al. (2021) | 489.75 | 14.32 | 587.31 | 16.61 |
| | PATE-GAN | Long et al. (2021) | 231.54 | 41.68 | 253.19 | 42.22 |
| $\varepsilon = 1$ | G-PATE | Long et al. (2021) | 153.38 | 58.80 | 214.78 | 58.12 |
| | DataLens | Wang et al. (2021) | 186.06 | 71.23 | 194.98 | 64.78 |
| | DP-MERF | Vinaroz et al. (2022)[8] | - | 80.7 | - | **73.9** |
| | DP-HP | Vinaroz et al. (2022) | - | **81.5** | - | 72.3 |
| | DPGAN | | 91.7 | 77.4 | 151.9 | 65.0 |
| | + large batches | (This work) | 66.1 | 73.7 | 153.2 | 66.6 |
| | + step schedule | | **56.2** | 80.1 | **121.8** | 68.0 |

Table 1: We gather previously reported results in the literature on the performance of various methods for labelled generation of MNIST and FashionMNIST. Note that *Reported In* refers to the source of the numerical result, not the originator of the approach. For downstream accuracy, we report the best accuracy among classifiers they use, and compare against our CNN classifier accuracy.

| Privacy | Method | Reported In | FID | Acc.(%) |
|---|---|---|---|---|
| $\varepsilon = \infty$ | Real data | (This work) | 1.1 | 96.6 |
| | GAN | | 31.5 | 91.6 |
| | DP-MERF | Cao et al. (2021) | 274.0 | 65 |
| $\varepsilon = 10$ | DP-Sinkhorn | Cao et al. (2021) | 189.5 | 76.3 |
| | DPGAN | (This work) | **166.8** | **83.8** |

Table 2: Comparison to state-of-the-art results on $32 \times 32$ CelebA-Gender, targeting $(10, 10^{-6}$-DP).

respectively), and employed large batches ($B = 1024$) and adaptive discriminator step frequency with threshold $d = 0.6$. Results are summarized in Table 2, example images are in Figure 11.

## 6 DISCUSSION AND RELATED WORK

**DP generative models.** The baseline DPGAN that employs a DPSGD-trained discriminator was introduced by Xie et al. (2018), and was subsequently studied in several works (Torzdehmahani et al., 2019; Beaulieu-Jones et al., 2019). Despite significant interest in the approach and numerous applications to various problems ($\approx 300$ citations as of November 2022), we were unable to find studies that explore the modifications we perform or uncover similar principles for improving training. Perhaps as a consequence, subsequent work has departed from this approach, examining alternative privatization schemes for GANS (Jordon et al., 2019; Long et al., 2021; Chen et al., 2020;

---

[7]We group per-class unconditional GANs together with conditional GANs under the DPGAN umbrella.
[8]These results are presented graphically in the paper. Exact numbers can be found in their code.

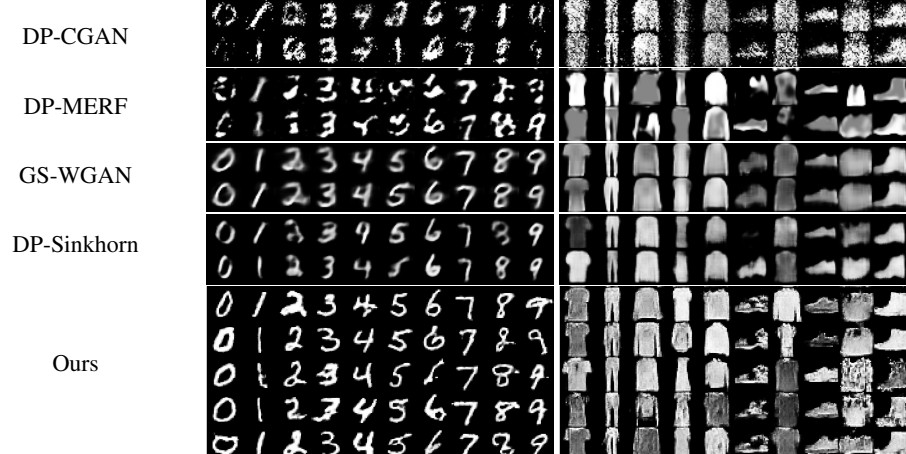

Figure 5: MNIST and FashionMNIST results at $(10, 10^{-5})$-DP for different methods. Images of other methods from (Cao et al., 2021).

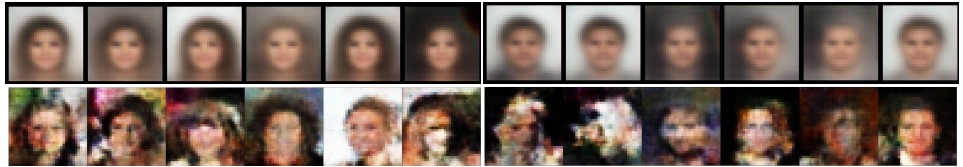

Figure 6: $32 \times 32$ CelebA-Gender at $(10, 10^{-6})$-DP. **Top:** DP-Sinkhorn. **Bottom:** Ours.

Wang et al., 2021). Contrary to their claims, our work shows that these privatization schemes do not outperform DPSGD. Other generative modelling frameworks have been applied to DP synthetic data including VAEs (Chen et al., 2018), maximum mean discrepancy (Harder et al., 2021; Vinaroz et al., 2022), Sinkhorn divergences (Cao et al., 2021), and normalizing flows (Waites & Cummings, 2021). We show that a well-tuned DPGAN competes with or outperforms these approaches.

**Custom approaches versus a well-tuned DPSGD.** An ongoing debate pertains to the best techniques and architectures for private ML. Roughly speaking, there are two schools of thought. One investigates novel architectures for privacy, which may be outperformed by more traditional approaches in the non-private setting. Some examples include Chen et al. (2018); Cao et al. (2021); Vinaroz et al. (2022), a variety of generative models specifically designed to be compatible with differential privacy. The other focuses on searching within the space of tried-and-tested methods that are understood to work well non-privately. Some examples include the works of De et al. (2022); Li et al. (2022), who demonstrate that, similar to the non-private setting, large-scale CNN and Transformer architectures can achieve state-of-the-art results for image classification and NLP tasks. The primary modifications to the pipeline are along the lines of changing the batch size, modifying the type of normalization layers, etc., most of which would be explored in a proper hyperparameter search in the non-private setting. Our work fits into the latter line: we show that novel generative models introduced for privacy can be outperformed by GANs trained with well-tuned DPSGD.

**Tabular data.** Our investigation focused on image datasets, while many important applications of private data generation involve *tabular data*. While Tao et al. (2021) find that private GAN-based approaches fail to preserve even basic statistics in these settings, we believe that our techniques may yield similar improvements.

# 7 CONCLUSION

Our most important contribution is to show that private GANs have been underrated by the research community, and can achieve state-of-the-art results with careful tuning. We hope and anticipate this will inspire the community to revisit private GANs, and quickly improve upon our results.

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

# A  GENERATED SAMPLES

We provide a few non-cherrypicked samples for MNIST and FashionMNIST at $\varepsilon = 10$ and $\varepsilon = 1$, as well as $32 \times 32$ CelebA-Gender at $\varepsilon = 10$.

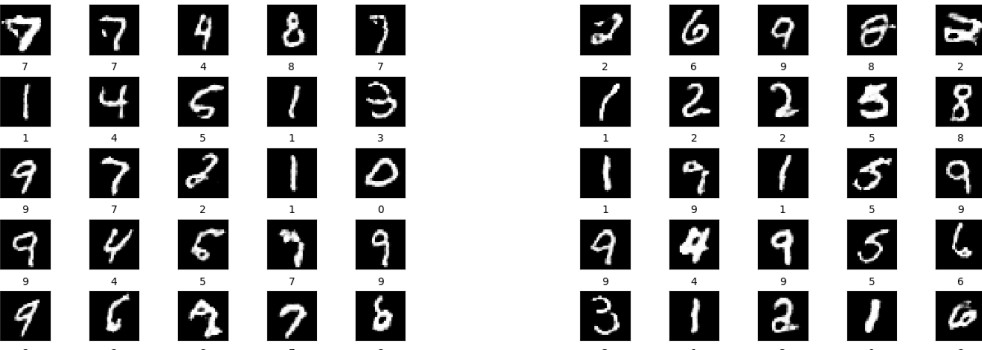

Figure 7: Some non-cherrypicked MNIST samples from our method, $\varepsilon = 10$.

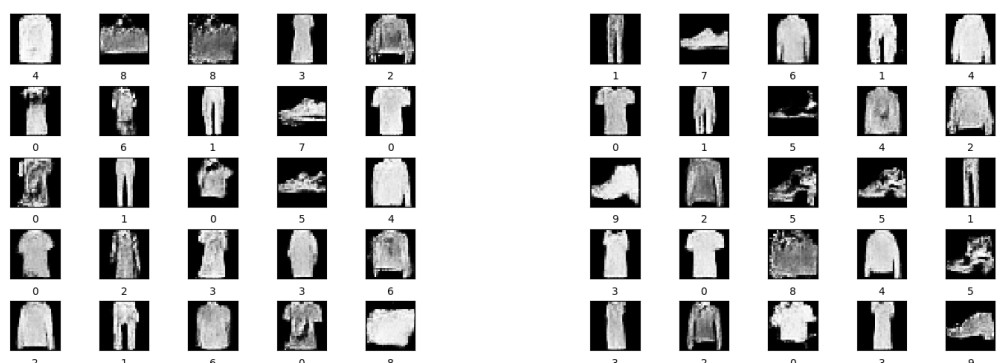

Figure 8: Some non-cherrypicked FashionMNIST samples from our method, $\varepsilon = 10$.

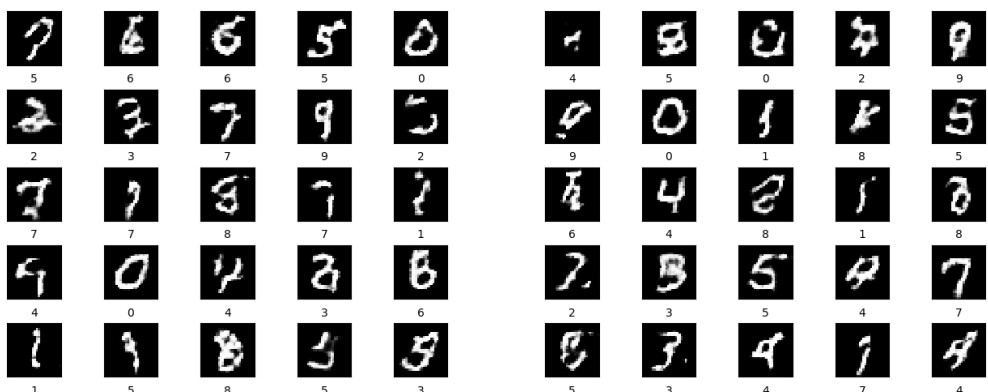

Figure 9: Some non-cherrypicked MNIST samples from our method, $\varepsilon = 1$.

# B  IMPLEMENTATION DETAILS

## B.1  MNIST AND FASHIONMNIST TRAINING RECIPE

For MNIST and FashionMNIST, we begin from an open source PyTorch implementation of DC-GAN (Radford et al., 2016) (available at this link) that performs well non-privately, and copy their

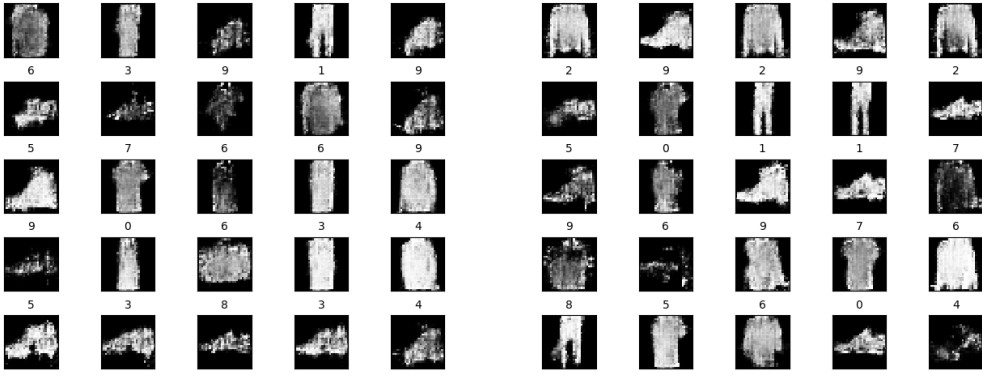

Figure 10: Some non-cherrypicked FashionMNIST samples from our method, $\varepsilon = 1$.

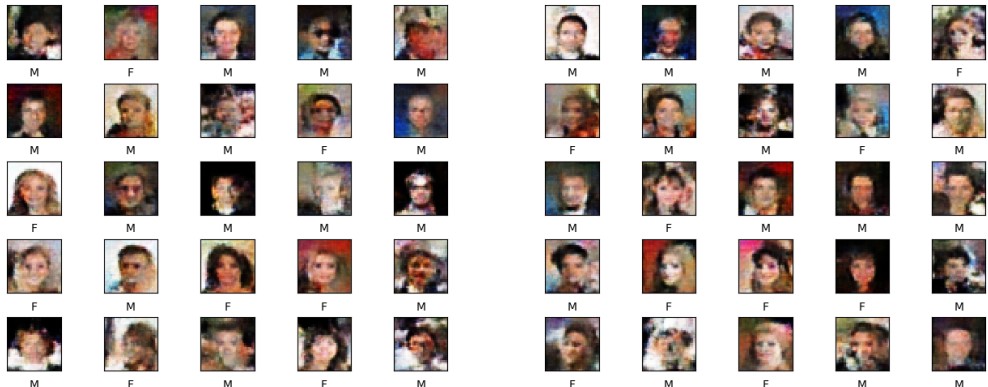

Figure 11: Some non-cherrypicked CelebA samples from our method, $\varepsilon = 10$.

training recipe. This includes: batch size $B = 128$, the Adam optimizer (Kingma & Ba, 2015) with parameters ($\alpha = 0.0002, \beta_1 = 0.5, \beta_2 = 0.999$) for both $\mathcal{G}$ and $\mathcal{D}$, the non-saturating GAN loss (Goodfellow et al., 2014), and a 5-layer fully convolutional architecture with width parameter $d = 128$.

To adapt it to our purposes, we make three architectural modifications: in both $\mathcal{G}$ and $\mathcal{D}$ we (1) remove all BatchNorm layers (which are not compatible with DPSGD); (2) add label embedding layers to enable labelled generation; and (3) adjust convolutional/transpose convolutional stride lengths and kernel sizes as well as remove the last layer, in order to process $1 \times 28 \times 28$ images without having to resize. Finally, we remove their custom weight initialization, opting for PyTorch defaults.

Our baseline non-private GANs are trained for $45000$ steps. We train our non-private GANs with poisson sampling as well: for each step of discriminator training, we sample real examples by including each element of our dataset independently with probability $B/n$, where $n$ is the size of our dataset. We then add $B$ fake examples sampled from $\mathcal{G}$ to form our fake/real combined batch.

### B.2 LARGE BATCH SIZE HYPERPARAMETER SEARCH

We scale up batch sizes, considering $B \in \{64, 128, 512, 2048\}$, and search for the optimal noise scale $\sigma$ and $n_{\mathcal{D}}$. For $B = 128$ targeting $\varepsilon = 10$, we search over three noise scales, $\Sigma_{B=128}^{\varepsilon=10} = \{0.6, 1.0, 1.4\}$. We choose candidate noise scales for other batch sizes as follows: when considering a batch size $128k$, we search over $\Sigma_{B=128k}^{\varepsilon=10} := \{\sqrt{k} \cdot \sigma : \sigma \in \Sigma_{B=128}^{\varepsilon=10}\}$. We also target the high privacy ($\varepsilon = 1$) regime. For $\varepsilon = 1$, we multiply all noise scales by 5, $\Sigma_B^{\varepsilon=1} = \{5\sigma : \sigma \in \Sigma_B^{\varepsilon=10}\}$. We search over a grid $n_D \in \{1, 2, 5, 10, 20, 50, 100, 200, 500\}$. Due to compute limitations, we omit some values that we are confident will fail (e.g., trying $n_{\mathcal{D}} = 1$ when mode collapse occurs for $n_{\mathcal{D}} = 5$).

## C ADDITIONAL DISCUSSION

**GANhacks.** Guidance in the non-private setting (tip 14 of Chintala et al. (2016)) prescribes to train the discriminator for more steps in the presence of noise (a regularization approach used in non-private GANs). This is the case for DP, and is our core strategy that yields the most significant gains in utility. We were not aware of this tip when we discovered this phenomenon, but it serves as validation of our finding. While Chintala et al. (2016) provides little elaboration, looking at further explorations of this principle in the non-private setting may offer guidance for improving DPGANs.

