# OpenReview forum: "Private GANs, Revisited"
_ICLR.cc/2023/Conference — Submitted to ICLR 2023_

### Official Review · Reviewer_hSom · 2022-10-19

**Confidence:** 3
**Correctness:** 4
**Technical Novelty And Significance:** 2
**Empirical Novelty And Significance:** 2
**Recommendation:** 5

**Clarity, Quality, Novelty And Reproducibility:**

The paper is written in a clear and easy to follow fashion, the work might be important to the community although the novelty of the work is not substantial. The work could be easily be reproduce.

**Details Of Ethics Concerns:**

no ethics concerns

**Strength And Weaknesses:**

### Strength
The paper provides a simple yet effective method to improve the quality of the generated data of DPGAN. The experiments show significant improvement in the tested benchmarks. The authors discuss in length about whey more steps are needed in the case of DPGAN. In addition, the authors discuss the role of the batch size in DPGAN settings and the additional benefits of scheduling the discriminator frequency update (start low then increase).

### Weaknesses
The novelty of the work is marginal. The method work best at \epsilon = 10, but on high privacy/ low \epsilon regime sometime the results are even worse.

**Summary Of The Paper:**

The authors revisit the DPGAN paper and state that the added noise to discriminator training disrupts the balance between the generator and discriminator training. They show that by tuning the number of update steps taking by the discriminator for every generator update step, specifically, taking more steps significantly improves the results. They show that for MNIST \at epsilon = 10 their private GAN FID goes from 48.4  to 13.0 and the downstream accuracy of the classifier goes up from 83.2% to 95.0%.

**Summary Of The Review:**

The paper present a very slight modification, a focus on a parameter change in the DPGAN method. It propose to increase the discriminator update frequency when noise is introduced to the discriminator (as in the DPGAN training case). The paper discuss the motivation to their method at length. Finally, the authors show inconclusive experiments, where the method works for some regime of the noise level.

---

> ### Author Response · Authors · 2022-11-24
> **Response to reviewer hSom**
>
> Thank you for the thoughtful response and comments on our work.
>
> We are glad the reviewer appreciates the significant improvements we obtain on standard benchmarks, and furthermore believes our work might be important to the community – indeed, we believe this is the most valuable aspect of our work.
>
> We emphasize that **the core contribution of our work is our novel and significant empirical findings**. We have revised our writing – in particular, the introduction – to make this more apparent. Our contributions, presented in the paper in order of importance are:
>
> 1. **(Most important): Falsifying the predominant view** in the literature (corroborated by numerous studies presented at top venues: NeurIPS [1, 2, 3], ICLR [4], ICML [5], AISTATS [6], CCS [7]) **that DPSGD-trained GANs are ineffective**, and are dramatically outperformed by various custom approaches designed with privacy in mind (see in Table 1, the large performance gaps between previously reported DPGAN results and the various custom approaches).
>
> 2. **(2nd most important)**: An empirical analysis of the phenomenon that increasing discriminator step frequency improves DPGANs, culminating in a proposed explanation.
>
> 3. **(Least important)**: Provide additional validation of our proposed explanation, by showing that making modifications to training based on the consequences of our explanation, indeed leads to better DPGANs.
>
> In the following, we discuss specific concerns raised by the reviewer.
>
> **On novelty:** As described above, our most important contribution is a **new empirical finding that contradicts the predominant view in the DP synthetic data literature that DPGANs are ineffective**. This is novel. These findings surprised us – leading us to empirically analyze the phenomenon, which culminated in an explanation motivating some simple and highly effective modifications to DPGANs. This renewed understanding of DPGAN training is also novel. We stress that the simplicity of our solutions *should not be conflated* with a lack of novelty in our findings and analysis.
>
> We note that, *a priori*, given the state of the literature, it is not clear at all that DPGANs can perform well, and moreover, be competitive with state-of-the-art approaches *with any modifications* – let alone the simple ones we show are sufficient in this paper. Examining any of the previously cited papers proposing new methods [1-7] will have DPGANs falling far short of state-of-the-art approaches.
>
> This is not for lack of trying: there has been significant interest in the original DPGAN paper [8] and numerous applications to various problems ($\approx 300$ citations as of November 2022). Despite this, we were unable to find any study that explores the modifications we perform or uncover similar principles for improving training.
>
> We believe that our finding –  achieving state-of-the-art approaches with the most simple and natural baseline (essentially, an instantiation of DPSGD) – is a significantly more novel contribution to the literature than if we had achieved similar results with yet another new custom method, in the same vein of [1-7]. Such a result would only serve to corroborate our existing *flawed* understanding of DP synthetic data generation (custom approaches work better). Our work suggests a change in paradigm for this area: *let’s push the limits of DPSGD on approaches that have been shown to be effective non-privately*.

---

> > ### Author Response · Authors · 2022-11-24
> > **Response to reviewer hSom**
> >
> > **On the strength of our experimental results:**
> >
> > Our goal was to demonstrate that our simple modifications dramatically improve the baseline, and are competitive with the state-of-the-art. While further engineering may lead to more improvements in various cases, we do not believe it is central to our message.
> >
> > Overall, we should not expect such simple modifications to a baseline method to uniformly outperform all custom-designed state-of-the-art approaches, across all metrics, privacy levels, and datasets (although we do in terms of FID, even in the low $\varepsilon$ regime).
> >
> > Note also that state-of-the-art approaches do not meet this criteria: certain approaches (DP-Sinkhorn [3], GS-WGAN [1]) only report favourable results for $\varepsilon=10$; on the other hand DP-MERF [6] and DP-HP [5] do not significantly improve at $\varepsilon=10$.
> >
> > If we take a look at Table 1 more granuarly: we can make two observations. In the $\varepsilon=1$ case, we are only outperformed by DP-MERF and DP-MERF, two approaches designed for the high privacy regime that do not see much improvement when allocated more privacy budget (or alternatively, when there is more data available). Note how DP-MERF only sees $\approx 1.5\\%$ accuracy improvement when going from $\varepsilon=1$ to $\varepsilon=10$.
> >
> > Table 1 in our paper also shows that, in particular, before applying the adaptive discriminator step frequencies and larger batch sizes, simply choosing an optimal setting of $n_{\mathcal d}$ in our baseline DPGAN (amounting to a 1 line of code change) outperforms all previous alternative GAN privatization schemes (PATE-GAN, GS-WGAN, G-PATE, DataLens) *uniformly* across all metrics, datasets, and privacy levels.
> >
> > Our revised draft has many more pictures of samples and visual comparisons with previous methods, including on more realistic image datasets (CelebA). We believe that these images lend credence to quality of our generation results.
> >
> > **Final comments:**
> >
> > Ultimately, the case we would like to make for our paper is on the significance of our findings. While the core modifications are simple, they give dramatic improvements to the utility of DPGANs. We view the simplicity as a benefit. This will lead to these ideas actually affecting the practice of differentially private machine learning. Furthermore, by falsifying the predominant view on DPGANs in the literature, our work challenges conventional wisdom regarding how to create private generative models, pointing toward a new direction for the area.
> >
> > The focus of our study is not on proposing new methods, but rather to revisit and clarify our understandings of existing ones. In at least partial agreement with the reviewer, we believe this kind of work is also of value to the ICLR community.
> >
> > Finally, as evidence for impact: we note that shortly after the posting of this manuscript online, an arxiv preprint (https://arxiv.org/abs/2205.12900) from authors working in this area have revised their experiments with DPGANs, citing our findings.

---

> > > ### Author Response · Authors · 2022-11-24
> > > **References**
> > >
> > > **References**
> > >
> > > [1] Dingfan Chen, Tribhuvanesh Orekondy, and Mario Fritz. GS-WGAN: A gradient-sanitized approach for learning differentially private generators. In *Advances in Neural Information Processing Systems 33 (NeurIPS’20)*, 2020.
> > >
> > > [2] Yunhui Long, Boxin Wang, Zhuolin Yang, Bhavya Kailkhura, Aston Zhang, Carl Gunter, and Bo Li. G-PATE: Scalable differentially private data generator via private aggregation of teacher discriminators. In *Advances in Neural Information Processing Systems 34 (NeurIPS’21)*, 2021.
> > >
> > > [3] Tianshi Cao, Alex Bie, Arash Vahdat, Sanja Fidler, and Karsten Kreis. Don’t generate me: Training differentially private generative models with Sinkhorn divergence. In *Advances in Neural Information Processing Systems 34 (NeurIPS’21)*, 2021.
> > >
> > > [4] James Jordon, Jinsung Yoon, and Mihaela van der Schaar. PATE-GAN: Generating synthetic data with differential privacy guarantees. In *7th International Conference on Learning Representations (ICLR’19)*, 2019.
> > >
> > > [5] Margarita Vinaroz, Mohammad-Amin Charusaie, Frederik Harder, Kamil Adamczewski, and
> > > Mi Jung Park. Hermite polynomial features for private data generation. In *Proceedings of the
> > > 39th International Conference on Machine Learning (ICML’22)*, 2022.
> > >
> > > [6] Frederik Harder, Kamil Adamczewski, and Mijung Park. DP-MERF: Differentially private mean embeddings with random features for practical privacy-preserving data generation. In *24th International Conference on Artificial Intelligence and Statistics (AISTATS’21)*, 2021.
> > >
> > > [7] Boxin Wang, Fan Wu, Yunhui Long, Luka Rimanic, Ce Zhang, and Bo Li. DataLens: Scalable
> > > privacy preserving training via gradient compression and aggregation. In *CCS’21: 2021 ACM
> > > SIGSAC Conference on Computer and Communications Security*, 2021.
> > >
> > > [8] Liyang Xie, Kaixiang Lin, Shu Wang, Fei Wang, and Jiayu Zhou. Differentially private generative adversarial network. *CoRR*, abs/1802.06739, 2018.

---

### Official Review · Reviewer_dASH · 2022-10-25

**Confidence:** 2
**Clarity, Quality, Novelty And Reproducibility:** Some details are missing, code is not…
**Correctness:** 3
**Technical Novelty And Significance:** 2
**Empirical Novelty And Significance:** 2
**Recommendation:** 3

**Strength And Weaknesses:**

Strength

1. In the literature of differential privacy gans, it seems like the utilization of more recent architectures and training tricks (e.g. StyleGAN2,3, …) for higher quality samples are mostly unexplored. This paper provides a good point that these “tricks” could largely boost the performance of private gans.

Weakness

1. From my perspective, this work is with limited novelty. Since larger batch size and take more discriminator steps has been studied widely in normal gan papers. It seems like natural attempts to try these tricks on private gans.
2. Several missing details. For example, how does (10, 10^-5)-DP calculated from B=128,sigma=1,C=1,T=450000 (in section 3.1)? Please provide the formal equation on calculating this. In addition, noise scales, n_D, and batch size also affect privacy, in section 5.1, how can you targeting the same \epsilon with different values of these?

**Summary Of The Paper:**

This paper provides two empirical findings on how to train differential private gans: larger batch size, and more discriminator steps. Experimental results show superior performance against existing baselines.

**Summary Of The Review:**

Overall, this paper is a practical guide for training private gans. However the findings are empirical and rather trivial. In addition, some details are missing.

---

> ### Author Response · Authors · 2022-11-23
> **Response to reviewer dASH**
>
> Thank you for the thoughtful response and comments on our work.
>
> First, we would like to emphasize that **the core contribution of our work is our novel and significant empirical findings**. We have revised our writing – in particular, the introduction – to make this more apparent. Our contributions, presented in the paper in order of importance are:
>
> 1. **(Most important): Falsifying the predominant view** in the literature (corroborated by numerous studies presented at top venues: NeurIPS [1, 2, 3], ICLR [4], ICML [5], AISTATS [6], CCS [7]) **that DPSGD-trained GANs are ineffective**, and are dramatically outperformed by various custom approaches designed with privacy in mind (see in Table 1, the large performance gaps between previously reported DPGAN results and the various custom approaches).
>
> 2. **(2nd most important):** An empirical analysis of the phenomenon that increasing discriminator step frequency improves DPGANs, culminating in a proposed explanation.
>
> 3. **(Least important):** Provide additional validation of our proposed explanation, by showing that making modifications to training based on the consequences of our explanation, indeed leads to better DPGANs.
>
> The reviewer’s primary critique of our work is that it lacks novelty. We hope that our response can offer compelling evidence against this view. As discussed above, our most important contribution is a **new empirical finding that contradicts the predominant view in the DP synthetic data literature that DPGANs are ineffective**. This is novel. These findings surprised us – leading us to empirically analyze the phenomenon, which culminated in an explanation motivating some simple and highly effective modifications to DPGANs. We stress that the simplicity of these solutions *should not be conflated* with a lack of novelty in our findings and analysis.
>
> In the following, we discuss specific concerns raised by the reviewer.
>
> > *1. From my perspective, this work is with limited novelty. Since larger batch size and take more discriminator steps has been studied widely in normal gan papers. It seems like natural attempts to try these tricks on private gans.*
>
> **On novelty:** First, we note that *a priori*, given the state of the literature, it is not clear at all that DPGANs can perform well, and moreover, be competitive with state-of-the-art approaches *with any modifications* – let alone the simple ones we show are sufficient in this paper.
>
> Now suppose even that we are somehow convinced that DPGANs can be competitive with state-of-the-art approaches (flying in the face of the findings of numerous recent works published at top venues). We argue that, *a priori*, it is not obvious what changes should be made to improve private GANs. The reviewer points to larger batch sizes and a higher discriminator step frequency to be “natural attempts”. We disagree, and believe that these modifications only appear natural *ex-post-facto* – in light of our findings and proposed explanation.
>
> First, we note that non-private GANs typically do not take advantage of large batch sizes and increased discriminator step frequencies. For example, StyleGAN2, as mentioned by the reviewer, uses a batch size of 32 [8] and takes 1 discriminator step per generator step [9]. This is different from our private GANs, which show improvement from using batch sizes as large as 2048 and step frequencies as large as 200. To corroborate this claim, we examine the effect of these modifications on the non-private version of the baseline GANs from Section 3. Results are pictured in Table D and E. In the non-private setting, these modifications do not result in significant gains.
>
> | **Batch Size** | **FID** | **Accuracy** |
> |:---|---:|---:|
> | 128   |   3.2 |        96.8 |
> | 512   |   3.1 |        97.4 |
> | 2048 |   3.2 |        97.5 |
>
> Table D: Effect of batch size on non-private performance for MNIST. We fix the number of steps, so the $B=2048$ goes through $20\times$ as many passes through the data.
>
> | **Discriminator step frequency ($n_{\mathcal D}$)** | **FID** | **Accuracy** |
> |:---|---:|---:|
> | 1   |   3.2 |        96.8 |
> | 10   |   8.3 |        91.3 |
>
> Table E: Effect of discriminator step frequency on non-private performance for MNIST. We fix the number of generator steps, so $n_{\mathcal D} = 10$ goes through $10\times$ as many passes through the data.
>
> Secondly, we note that despite significant interest in the original private GAN paper [10] and numerous applications to various problems ($\approx 300$ citations as of November 2022), we were unable to find any study that explores the modifications we perform or uncover similar principles for improving training. This supports our claim that these modifications are non-obvious.

---

> > ### Author Response · Authors · 2022-11-23
> > **Response to reviewer dASH**
> >
> > > *2. Several missing details. For example, how does (10, 10^-5)-DP calculated from B=128,sigma=1,C=1,T=450000 (in section 3.1)? Please provide the formal equation on calculating this.*
> >
> > **On privacy cost calculation details:** As described in our manuscript, we use the RDP privacy accounting in [11], as implemented in Opacus [12]. The privacy accounting used here follows the standard DPSGD analysis [13]:
> > - Computing the RDP privacy cost of the subsampled gaussian mechanism (which is what’s used for each discriminator update)
> > - Applying RDP composition over all timesteps, to get a total RDP privacy cost for training
> > - Converting from the RDP guarantee to an $(\varepsilon,\delta)$ guarantee.
> >
> > We follow the standard practice in the differentially private machine learning literature of reporting the privacy accounting method and implementation used, along with the parameters necessary for calculation (see for example, the level of detail in papers appearing previously at ICLR [14,15,16]). We believe these details are sufficient to reproduce our work.
> >
> > > *In addition, noise scales, n_D, and batch size also affect privacy, in section 5.1, how can you targeting the same \epsilon with different values of these?*
> >
> > First we would like to clarify that $n_\mathcal{D}$, the number of discriminator steps taken between generator steps, does not affect the total privacy cost. When we compare different values of $n_{\mathcal D}$, we fix the total number of discriminator steps, only varying the length of interval between discriminator steps, that we take generator steps. Note that the number of generator steps taken does not affect privacy.
> >
> > The privacy cost of DPSGD is entirely determined by the batch size $B$, noise scale $\sigma$, and number of steps $T$. Fixing $B$ and $\sigma$, we find the maximal number of steps $T$ within our targeted $\varepsilon$.
> >
> > We hope that our comments here address the technical concerns regarding reproducibility and correctness raised by the reviewer. We would be happy to clarify any additional concerns.
> >
> > **Final comments:**
> >
> > Ultimately, the case we would like to make for our paper is on the significance of our findings. While the core modifications are simple, they give dramatic improvements to the utility of DPGANs. We view the simplicity as a benefit. This will lead to these ideas actually affecting the practice of differentially private machine learning. Furthermore, by falsifying the predominant view on DPGANs in the literature, our work challenges conventional wisdom regarding how to create private generative models, pointing toward a new direction for the area.
> >
> > The focus of our study is not on proposing new methods, but rather to revisit and clarify our understandings of existing ones. We believe this kind of work is also of value to the ICLR community.
> >
> > Table 1 in our paper shows that, in particular, before applying the adaptive discriminator step frequencies and larger batch sizes, simply choosing an optimal setting of $n_{\mathcal D}$ in our baseline DPGAN (amounting to a 1 line of code change) outperforms all previous alternative GAN privatization schemes (PATE-GAN, GS-WGAN, G-PATE, DataLens, all much more complex than plain DPSGD) uniformly across all metrics, datasets, and privacy levels. We conjecture it is perhaps the focus on proposing novel approaches, rather than on analyzing and understanding existing ones (figuring out when and why they succeed and fail), that has led to such dramatic misconceptions in the literature.
> >
> > We thank the reviewer for their suggestions and for engaging with our work. We hope that our comments here and our revisions helped address the reviewer’s concerns.

---

> > > ### Author Response · Authors · 2022-11-23
> > > **References**
> > >
> > > **References**
> > >
> > > References
> > > [1] Dingfan Chen, Tribhuvanesh Orekondy, and Mario Fritz. GS-WGAN: A gradient-sanitized approach for learning differentially private generators. In *Advances in Neural Information Processing Systems 33 (NeurIPS’20)*, 2020.
> > >
> > > [2] Yunhui Long, Boxin Wang, Zhuolin Yang, Bhavya Kailkhura, Aston Zhang, Carl Gunter, and Bo Li. G-PATE: Scalable differentially private data generator via private aggregation of teacher discriminators. In *Advances in Neural Information Processing Systems 34 (NeurIPS’21)*, 2021.
> > >
> > > [3] Tianshi Cao, Alex Bie, Arash Vahdat, Sanja Fidler, and Karsten Kreis. Don’t generate me: Training differentially private generative models with Sinkhorn divergence. In *Advances in Neural Information Processing Systems 34 (NeurIPS’21)*, 2021.
> > >
> > > [4] James Jordon, Jinsung Yoon, and Mihaela van der Schaar. PATE-GAN: Generating synthetic data with differential privacy guarantees. In *7th International Conference on Learning Representations (ICLR’19)*, 2019.
> > >
> > > [5] Margarita Vinaroz, Mohammad-Amin Charusaie, Frederik Harder, Kamil Adamczewski, and
> > > Mi Jung Park. Hermite polynomial features for private data generation. In *Proceedings of the
> > > 39th International Conference on Machine Learning (ICML’22)*, 2022.
> > >
> > > [6] Frederik Harder, Kamil Adamczewski, and Mijung Park. DP-MERF: Differentially private mean embeddings with random features for practical privacy-preserving data generation. In *24th International Conference on Artificial Intelligence and Statistics (AISTATS’21)*, 2021.
> > >
> > > [7] Boxin Wang, Fan Wu, Yunhui Long, Luka Rimanic, Ce Zhang, and Bo Li. DataLens: Scalable
> > > privacy preserving training via gradient compression and aggregation. In *CCS’21: 2021 ACM
> > > SIGSAC Conference on Computer and Communications Security*, 2021.
> > >
> > > [8] Issue on StyleGAN2-PyTorch repository. Github. https://github.com/rosinality/stylegan2-pytorch/issues/152.
> > >
> > > [9] StyleGAN2 code. Github. https://github.com/NVlabs/stylegan2/blob/master/training/training_loop.py.
> > >
> > > [10] Liyang Xie, Kaixiang Lin, Shu Wang, Fei Wang, and Jiayu Zhou. Differentially private generative adversarial network. *CoRR*, abs/1802.06739, 2018.
> > >
> > > [11] Ilya Mironov, Kunal Talwar, and Li Zhang. Renyi differential privacy of the sampled gaussian mechanism. *CoRR*, abs/1908.10530, 2019.
> > >
> > > [12] Ashkan Yousefpour, Igor Shilov, Alexandre Sablayrolles, Davide Testuggine, Karthik Prasad, Mani Malek, John Nguyen, Sayan Gosh, Akash Bharadwaj, Jessica Zhao, Graham Cormode, and Ilya Mironov. Opacus: User-friendly differential privacy library in PyTorch. *CoRR*, abs/2109.12298, 2021.
> > >
> > > [13] Martin Abadi, Andy Chu, Ian Goodfellow, H. Brendan McMahan, Ilya Mironov, Kunal Talwar, and Li Zhang. Deep learning with differential privacy. In *CCS’16: 2016 ACM SIGSAC Conference on Computer and Communications Security*, 2016.
> > >
> > > [14] Florian Tramer and Dan Boneh. Differentially private learning needs better features (or much more data). In *9th International Conference on Learning Representations (ICLR’21)*, 2021.
> > >
> > > [15] Xuechen Li, Florian Tramer, Percy Liang, and Tatsunori Hashimoto. Large language models can be strong differentially private learners. In *10th International Conference on Learning Representations (ICLR’22)*, 2022.
> > >
> > > [16] Da Yu, Saurabh Naik, Arturs Backurs, Sivakanth Gopi, Huseyin A. Inan, Gautam Kamath, Janardhan Kulkarni, Yin Tat Lee, Andre Manoel, Lukas Wutschitz, Sergey Yekhanin, and Huishuai Zhang. Differentially private fine-tuning of language models. In *10th International Conference on Learning Representations (ICLR’22)*, 2022.

---

### Official Review · Reviewer_wVny · 2022-10-30

**Confidence:** 3
**Correctness:** 3
**Technical Novelty And Significance:** 2
**Empirical Novelty And Significance:** 4
**Recommendation:** 5

**Clarity, Quality, Novelty And Reproducibility:**

The clarity and quality are good in general with some issues to be improved.
This paper is novel in terms of empirical findings, while the proposed method is currently limited to an engineering technique.
The reproducibility is good with sufficient implementation details provided.

**Strength And Weaknesses:**

Strength:

1. The proposed techniques are very effective to achieve much better FID on MNIST and FashionMNIST.

2. The empirical explanation of balancing the discriminator and maximizing generator steps taken when discriminator accuracy is high, is interesting and inspirational.

Weaknesses:

1. Overall, I think the proposed method is currently limited to an engineering technique, which might not be mature enough. However, I do believe it is highly potential to develop a principled method with high impact along the current direction. For example, I would recommend further elaborate on the step scheduler to be more adaptive and include more ablation studies. The authors might be able to draw some inspirations from StyleGAN2-ADA for such paradigm.

2. The improvement of utility accuracy is not as significant as FID. It would also be better to include an experiment on some larger datasets such as CelebA.

3. The writing and the organization could be improved.
- Although fancy, the title is not very informative.
- Figure 1 and figure 2 share the same purpose.
- Section 3.3 is duplicate to the previous content.
- Section 5.1 and 5.2 are not well aligned with the theme of Section 5 and the results are not well linked to the discriminator accuracy.
- In Algorithm 1, some symbols are used without definition.
- In Table 1, "(This work)" is ambiguous.

**Summary Of The Paper:**

This paper proposes a simple strategy to improve DP-GANs by using more update steps for discriminator, adopting a step scheduler, and taking a larger batch size. The authors provide empirical findings to justify such improvement by linking generation quality with the discriminator accuracy though the training process. Experiments on MNIST and FashionMNIST show very promising results in terms of generation quality.

**Summary Of The Review:**

This paper has interesting empirical findings and promising results, while at current stage the proposed solution is not mature enough to meet the standard of ICLR. However, along its direction, this work is potential to have an impact in the future, probably with an elaborated method or some theoretical support.

---

> ### Author Response · Authors · 2022-11-23
> **Response to reviewer wVny**
>
> Thank you for the thoughtful response and detailed comments regarding our work.
>
> We are very glad the reviewer finds our generation results promising (we added many more pictures of our samples and compared them with pictures from other methods in the revised draft – please check them out!). We also appreciate that the reviewer finds our empirical findings to be interesting and significant – with potential for high impact in this area.
>
> First, we would like to emphasize that the **core contribution of our work is indeed our novel and significant empirical findings**. We have revised our writing – in particular, the introduction – to make this more apparent. Our contributions, presented in the paper in order of importance are:
>
> 1. **(Most important): Falsifying the predominant view** in the literature (corroborated by numerous studies presented at top venues: NeurIPS [1, 2, 3], ICLR [4], ICML [5], AISTATS [6], CCS [7]) **that DPSGD-trained GANs are ineffective**, and are dramatically outperformed by various custom approaches designed with privacy in mind (see in Table 1, the large performance gaps between previously reported DPGAN results and the various custom approaches).
>
> 2. **(2nd most important):** An empirical analysis of the phenomenon that increasing discriminator step frequency improves DPGANs, culminating in a proposed explanation.
>
> 3. **(Least important):** Provide additional validation of our proposed explanation, by showing that making modifications to training based on the consequences of our explanation, indeed leads to better DPGANs.
>
> The concerns of the reviewer focus primarily on our final contribution (Section 5), where we make modifications to DPGAN training based on the understanding we arrive at in Section 4. We would like to stress that this is not the most important aspect of our work – our primary focus is not to propose a new approach, but rather, to bring to light important empirical findings regarding DPGANs, and DP synthetic data generation in general that have been overlooked in the literature. We strongly believe that with our paper, the story on private synthetic data told by the literature looks quite different, and points towards a different direction for this research area.
>
> In the following, we discuss specific concerns raised by the reviewer.
>
> > *1. Overall, I think the proposed method is currently limited to an engineering technique, which might not be mature enough … I would recommend further elaborate on the step scheduler to be more adaptive and include more ablation studies.*
>
> **On the proposed method:** As in the point made above, the focus of our work is to present and analyze novel empirical findings – not on designing more elaborate/adaptive methods. In Section 5, we wanted to showcase that making simple training modifications based on the immediate consequences of our proposed explanation (“parity disruption from DP”) does in fact improve DPGANs, thereby validating our explanation. For this purpose, simple modifications that do not risk conflating many changes made at once are more valuable. Designing and experimenting with more elaborate and adaptive methods (with the hopes of scaling up to larger datasets) is exciting and we leave it to future work.
>
> **Ablations:** We present the results of some ablations on the effect of batch size and step schedule hyperparameters for MNIST at $\varepsilon=10$.
>
> | **Batch Size** | **FID** | **Accuracy** |
> |:---|---:|---:|
> | 64     |   25.2 |        92.0 |
> | 128   |   19.4 |        92.9 |
> | 512   |   14.1 |        94.2 |
> | 2048 |   13.2 |        94.3 |
>
> **Table A:** The effect of batch size on performance. Setting: no step schedule, reporting the best result among update frequencies and noise levels for each batch size.
>
> | Input accuracies to step scheduler | **FID** | **Accuracy** |
> |:---|---:|---:|
> | True discriminator accuracy (oracle)     |   10.8 |        93.9 |
> | Discriminator accuracy on fake samples only (proxy)  |   11.7 |       93.5|
>
> **Table B:** Comparing the use of true discriminator accuracy vs. using discriminator accuracy on fake samples only as a proxy for true accuracy. Note that the true accuracy is non-private; the focus of this experiment is to see how well the proxy approximates the oracle performance. Setting: *batch size* 2048, 200-step *grace period*, reporting the best result among discriminator accuracy thresholds $d$ for each setting.
>
>
> | **Discriminator Accuracy threshold $d$** | **FID** | **Accuracy** |
> |:---|---:|---:|
> | 0.60  |  20.8 | 92.4 |
> | 0.65  |  13.3 | 93.6 |
> | 0.70  |  11.7 | 93.5 |
> | 0.75  |  15.2 | 92.6 |
>
> **Table C:** The effect of the discriminator accuracy threshold $d$ on performance. Setting: *batch size* 2048, 200-step *grace period*.

---

> > ### Author Response · Authors · 2022-11-23
> > **Response to reviewer wVny**
> >
> > > *2. The improvement of utility accuracy is not as significant as FID. It would also be better to include an experiment on some larger datasets such as CelebA.*
> >
> > **On the utility accuracy of empirical results:** Compared to previously reported DPGAN results, our improvement in terms of utility accuracy is highly significant (for example, from 10% to 68% for FashionMNIST at $\varepsilon=1$) which validates the effectiveness of our modifications.
> >
> > For MNIST: at $\varepsilon=10$, our gains are significant in terms of utility accuracy, beating the previous state-of-the-art by over 10%. At $\varepsilon=1$, we beat all previous GAN-based approaches by almost 9%, and lose out to DP-MERF [6] and DP-HP [5], which are approaches specifically designed for the $\varepsilon=1$ regime, that do not see significant improvement when allocated larger privacy budgets.
> >
> > For FashionMNIST: our visual results and FID are quite good compared to prior work (see Figure 5). However we note that the non-private baseline GAN only achieves 80.4% accuracy – hence we believe that the low utility accuracies on FashionMNIST may be explained by the lack of jointly optimizing the GAN architecture and classifier training pipeline for FashionMNIST utility accuracy.
> >
> > Overall, it should be very surprising for our simple modifications to a baseline method to uniformly outperform all custom-designed state-of-the-art approaches, across all metrics, privacy levels, and datasets. Note also that state-of-the-art approaches do not meet this criteria: certain approaches (DP-Sinkhorn [3], GS-WGAN [1]) only report favourable results for $\varepsilon=10$; on the other hand DP-MERF [6] and DP-HP [5] do not significantly improve at $\varepsilon=10$. Our goal was to demonstrate that our simple modifications dramatically improve over the baseline, and are competitive with the state-of-the-art.
> >
> > **CelebA results:** In our revised draft, we include an experiment on CelebA. Table 2 shows that we outperform the previous state-of-the-art approach (DP-Sinkhorn [3]). Figure 6 compares samples drawn from our private GANs vs. samples drawn from DP-Sinkhorn. We note that qualitatively they are quite different: DP-Sinkhorn looks to be producing small variations on a “prototypical” male or female, whereas our results still resemble faces but are noticeably more diverse.
> >
> > We also note that on CelebA, our accuracy improvements are more significant than our FID improvements.
> >
> > > *3. The writing and the organization could be improved. Although fancy, the title is not very informative.*
> >
> > Yes the title could use some work. We are thinking of: *Private GANs are underrated*, which conveys our main message. We’re happy to take suggestions on the title.
> >
> > > *Figure 1 and figure 2 share the same purpose. Section 3.3 is duplicate to the previous content.*
> >
> > Since it is our main finding, we want to dedicate space to the results for MNIST in Figure 1 and FahsionMNIST in Figure 2. We cut parts of Section 3.3 and turned it into a paragraph rather than a whole section.
> >
> > > *Section 5.1 and 5.2 are not well aligned with the theme of Section 5 and the results are not well linked to the discriminator accuracy.*
> >
> > We respectfully disagree. The theme of Section 5 is to validate – along the lines of our explanation – that restoring parity by improving discriminator training (either through explicitly monitoring accuracy via the step schedule, or using large batches, which is known to improve accuracy in DP classification) improves generation.
> >
> > > *In Algorithm 1, some symbols are used without definition.*
> >
> > We fixed this.
> >
> > > *In Table 1, "(This work)" is ambiguous.*
> >
> > Can you describe what the ambiguity is? We can fix this also.
> >
> > **Final comments:**
> > We thank the reviewer for their suggestions and for engaging with our work. We hope that our comments here and our revisions helped address the reviewer’s concerns.
> >
> > We also hope that our revised manuscript emphasizes our intentions to clarify and gain understanding of existing methods, rather than forging onward with new approaches. We believe this kind of research is also significant and of interest to the ICLR community.
> >
> > Finally, as evidence for impact: we note that shortly after the posting of this manuscript online, an arxiv preprint (https://arxiv.org/abs/2205.12900) from authors working in this area have revised their experiments with DPGANs, citing our findings.

---

> > > ### Author Response · Authors · 2022-11-23
> > > **References**
> > >
> > > **References**
> > >
> > > [1] Dingfan Chen, Tribhuvanesh Orekondy, and Mario Fritz. GS-WGAN: A gradient-sanitized approach for learning differentially private generators. In *Advances in Neural Information Processing Systems 33 (NeurIPS’20)*, 2020.
> > >
> > > [2] Yunhui Long, Boxin Wang, Zhuolin Yang, Bhavya Kailkhura, Aston Zhang, Carl Gunter, and Bo Li. G-PATE: Scalable differentially private data generator via private aggregation of teacher discriminators. In *Advances in Neural Information Processing Systems 34 (NeurIPS’21)*, 2021.
> > >
> > > [3] Tianshi Cao, Alex Bie, Arash Vahdat, Sanja Fidler, and Karsten Kreis. Don’t generate me: Training differentially private generative models with Sinkhorn divergence. In *Advances in Neural Information Processing Systems 34 (NeurIPS’21)*, 2021.
> > >
> > > [4] James Jordon, Jinsung Yoon, and Mihaela van der Schaar. PATE-GAN: Generating synthetic data with differential privacy guarantees. In *7th International Conference on Learning Representations (ICLR’19)*, 2019.
> > >
> > > [5] Margarita Vinaroz, Mohammad-Amin Charusaie, Frederik Harder, Kamil Adamczewski, and
> > > Mi Jung Park. Hermite polynomial features for private data generation. In *Proceedings of the
> > > 39th International Conference on Machine Learning (ICML’22)*, 2022.
> > >
> > > [6] Frederik Harder, Kamil Adamczewski, and Mijung Park. DP-MERF: Differentially private mean embeddings with random features for practical privacy-preserving data generation. In *24th International Conference on Artificial Intelligence and Statistics (AISTATS’21)*, 2021.
> > >
> > > [7] Boxin Wang, Fan Wu, Yunhui Long, Luka Rimanic, Ce Zhang, and Bo Li. DataLens: Scalable
> > > privacy preserving training via gradient compression and aggregation. In *CCS’21: 2021 ACM
> > > SIGSAC Conference on Computer and Communications Security*, 2021.

---

### Decision · Program_Chairs · 2023-01-20

**Decision:**

Reject

**Justification For Why Not Higher Score:**

All reviewers thought the novelty of this paper is limited, as both the larger batch size and steps of tuning the discriminator are natural strategies in the literature. The overall approach is quite heuristic and lacks in-depth theoretical discussions.

**Justification For Why Not Lower Score:**

N/A

**Metareview: Summary, Strengths And Weaknesses:**

This paper studies strategies to train differential private gans, i.e., larger batch size and more steps to update the discriminator.

All reviewers recognize the strength of this paper is its simple yet effective way to improve the quality of the generated data of DPGAN. But they also share the same major concern on the limited novelty of this paper.